# Sensorineural Hearing Loss in Sjögren’s Syndrome

**DOI:** 10.3390/ijms231911181

**Published:** 2022-09-23

**Authors:** Yuko Okawa, Kenji Ihara

**Affiliations:** Department of Pediatrics, Oita University Faculty of Medicine, Oita 879-5593, Japan

**Keywords:** Sjögren’s syndrome, sensorineural hearing loss, high-frequency hearing loss, autoimmune mediated inner ear disease

## Abstract

Sjögren’s syndrome is a chronic autoimmune disease characterized by systemic dysfunction of exocrine glands, mainly the salivary and lachrymal glands. Sjögren’s syndrome consists of two forms: primary Sjögren’s syndrome, which is characterized by dry eyes and dry mouth without autoimmune diseases; and secondary Sjögren’s syndrome, which is characterized by symptoms associated with other autoimmune diseases, such as systemic lupus erythematosus. Disease severities vary considerably from mild glandular dryness to severe glandular involvement with numerous extraglandular and systemic features. Sensorineural hearing loss is sometimes observed in both primary and secondary Sjögren’s syndrome. This review article consists of (1) Pathology of Sjögren’s syndrome, (2) Clinical manifestation of Sjögren’s syndrome, (3) Autoimmune inner ear disease, (4) Histoanatomical features of the inner ear, (5) Immunological characteristics of the inner ear, (6) Pathophysiology of autoimmune inner ear disease, (7) Treatment for sensorineural hearing loss in Sjögren’s syndrome, and (8) Future direction. Finally, we introduce a recently developed disease model of salivary gland inflammation and discuss future expectations for the treatment of sensorineural hearing loss in Sjögren’s syndrome.

## 1. Pathology of Sjögren’s Syndrome (SS)

The pathophysiological features of SS are based on lymphocyte infiltration into the exocrine glands, mainly dry salivary and lacrimal glands, associated with extraglandular variable symptoms, such as a fever, lymph node swelling, or joint pain, induced by autoimmune reactions due to activated lymphocytes and released cytokines, chemokines, and immunoglobulins. The primary form of SS is not associated with other diseases, whereas the secondary form is complicated by other autoimmune diseases, such as systemic lupus erythematosus (SLE), rheumatoid arthritis (RA), scleroderma, and ulcerative colitis [1].

Among the extraglandular symptoms, neurological involvement of peripheral and central nerve systems is sometimes observed in patients with SS; however, hearing impairment is not well recognized in these patients, partly because of its general occurrence in a gradual manner as the natural consequence of aging. Because early treatment with steroids has been reported to successfully minimalize the progression of hearing loss, the early detection of sensorineural hearing loss (SNHS) in the SS patients is considered important [2].

We herein review the characteristic features of hearing loss in SS, focused on the pathophysiological and immunological mechanism as well as the treatment of SNHS. 

## 2. Clinical Manifestation of Sjögren’s Syndrome

### 2.1. Prevalence of Sjögren’s Syndrome

The autoimmune disease SS has worldwide distribution, and its clinical characteristics vary by ethnicity. The Big Data Sjögren Project Consortium conducted a multivariate logistic regression analysis involving 20 centers from 5 continents in January 2016 on an international multi-center registry designed in 2014. An analysis of this registry included 7748 women (93%) and 562 men (7%), with a mean age at diagnosis of primary SS of 53 years old, 95% of which listed ethnicity. This study demonstrated that SS was diagnosed on average 7 years earlier in black/African Americans compared with white patients. The female to male ratio was highest in Asian patients (27:1) and lowest in black/African American patients (7:1) [3].

### 2.2. Characteristic Symptoms and the Diagnosis of Sjögren’s Syndrome

The main symptoms of SS include ocular and oral dryness caused by autoimmunity in the exocrine glands (sicca symptoms) [1]. SS should be suspected in individuals with persistent sicca symptoms, parotid gland enlargement, and an unexplained increase in dental caries associated with specific serologic tests, such as positive findings for anti SS-A or SS-B antibody. The clinical diagnosis of SS is made in the presence of compatible clinical features and laboratory data with the exclusion of other diseases causing ocular and oral dryness [1]. 

## 3. Autoimmune Inner Ear Disease (AIED) 

The function of the inner ear is to convert sound information transmitted as vibrations into electrical signals and transmit them to the brain via the cochlear nerve. Sound enters the outer ear, reaches the eardrum in the middle ear, and travels to the inner ear via the ossicles attached to the eardrum. The inner ear consists of the cochlea, responsible for hearing, and the vestibule, responsible for balance. Vibration of the auditory ossicles causes the lymph fluid to vibrate, and the hair cells convert the vibrations into electrical signals that are transmitted to the cochlear nerve. Hair cells line the inside of the cochlea, and the frequencies assigned to each hair cell depend on where they are located. Hearing loss occurs when these hair cells are damaged by immune response (Figure 1). 

AIED is an inner ear dysfunction characterized by progressive and fluctuating tinnitus, vertigo, and SNHL. No definite, widely accepted markers for the diagnosis of AIED have been established, but the clinical diagnostic criteria are approximately defined as follows: (a) progressive, bilateral SNHL of ≥30 dB at ≥1 frequency; and (b) SNHL determined to be idiopathic based on a clinical evaluation, blood examinations, and magnetic resonance imaging (MRI) or computed tomography (CT) findings [4].

In a previous study, almost 30% of patients with AIED had coexisting systemic autoimmune disease, such as SLE, RA, or SS [5,6]. SNHL was the most common otologic symptom found in patients with SLE, with a wide range of prevalence from 6% to 70% depending on the methods and timing of the tests [7,8,9,10,11,12,13,14,15,16,17,18,19]. Hearing loss is mainly in the high frequencies, mimicking the typical presbycusis pattern as slowly progressive, but also can show acute or rapid progression, including in the low and middle frequencies. In addition to hearing loss, other audiovestibular symptoms, such as tinnitus and vertigo, are also often associated with SNHL. The investigators also speculated a relationship between the presence of anticardiolipin antibodies and SNHL, which is often seen in SLE patients, and sudden hearing loss [20,21,22]. RA is another well-known chronic inflammatory disease that affects 1% of the population, and the prevalence of associated SNHL ranges from 25% to 72% [23,24,25,26]. 

Regarding primary SS, it was estimated that approximately one-quarter of patients suffer from high-frequency hearing loss due to cochlear dysfunction as detected by impedance audiometry or auditory brainstem procedures [27]. In an extended study of SNHL in autoimmune disorders, the range of 500–8000 Hz was disturbed in 22–70% of SS patients, depending on the age, while hearing loss at higher frequencies was remarkably high, being almost 100% among SS patients for 10,000–16,000 Hz [28,29,30]. Table 1 shows the details of previous reports of SNHL in SS, SLE, and RA from 1997 to 2019. Most SS patients suffered from SNHL at very high frequencies above 9000 Hz. As intermediate frequencies from 500 to 2000 Hz are generally used for daily conversation, hearing difficulty is not often recognized for SS patients in their 40s or 50s. It is thus speculated that the ratio of bilateral SNHL with high frequencies may be much higher than reported, indicating the need for broad-spectrum audiometry. 

## 4. Histoanatomical Features of the Inner Ear [34]

The cochlea is the part of the inner ear involved in hearing, consisting of a base plate between the vestibular floor and the scala tympani floor. The movement of the stapes is transmitted as waves propagating through the inner ear fluid at 2 and 1/2 turns of the cochlea. The envelope signal shows the amplitude distribution of the entire basal lamina corresponding to each frequency. Frequency-specific motion of the organ of Corti causes depolarization of inner and outer hair cells by bending stereocilia. High-frequency sounds stimulate corresponding ganglion cells in the base of the cochlea, whereas low-frequency sounds stimulate those in the apex. Electrical impulses are then transmitted through the auditory nerve to the brain, which edits and perceives the transmitted information as complex sounds. 

The sound enters through the entrance of the cochlea, so hair cells near the entrance are heavily exposed to sound at all frequencies and wear out over time. Age-related hearing loss includes sounds mainly at high frequencies because of damage to hair cells on the inlet side of the affected cochlea [35]. The damaged hair cells do not regenerate, resulting in permanent high-frequency hearing loss. The damage to cochlear hair cells in SS patients is supposed be a similar or accelerated type of age-related hearing loss.

## 5. Immunological Characteristics in the Inner Ear 

The inner ear is separated by a blood/labyrinth barrier and is considered to be immunologically inactive, just like the brain [36,37]. However, it has been found to contain numerous resident macrophages, and the inner ear immune system communicates with the systemic immune system to maintain homeostasis [38]. The endolymphatic sac (ES) is part of the membranous labyrinth and is located in the dural duplication near the petrous bone and cerebellum. The ES is connected to the rest of the inner ear by filamentous endolymphatic vessels that travel to the ES in bone channels called vestibular aqueducts (VAs). VAs and the ES control homeostasis of the endolymph surrounding sensory hair cells. The antigens reached in the inner ear are captured and eliminated by the immune cell mechanism in the ES. This helps prevent inflammation near fragile structures of the inner ear [39].

## 6. Pathophysiology of Autoimmune Inner Ear Disease

AIED is an unusual form of progressive non-age-related SNHL and sometimes associated with vertigo. It occurs in both ears with cochlear and vestibular symptoms that progress over time and affect both ears. 

The following immunologic mechanisms are presumed to be involved in the pathophysiology of inner ear dysfunction due to autoimmune diseases:(a)Contribution of systemic circulating antibodies against inner ear antigens leading to antibody-dependent cell-mediated cytotoxicity with the activation of the complement system [40]. For example, IgG4-related disease caused inner ear involvement with cochlear ossification, and the presence of inner ear antibody was previously reported as the cause of inner ear dysfunction [41].(b)FcR-mediated activation of inflammatory reaction by immune complex deposition [42]. The immune complex deposition led to vasculitis of the inner ear vessels and determined atrophy of the stria vascularis. In a mouse model (by Iwai), abnormal helper T lymphocytes induced autoantibody production from B lymphocytes, and the resulting antigen–antibody complexes deposited in cochlear striae, causing hearing loss [43]. It is suspected that deposition of immune complexes reduced the blood flow in caliber of the auditory arteries and induced damage of the hair cells or the spiral ganglion by reactive oxygen species (ROS).(c)Autoinflammatory microvasculitis in labyrinthine arteries. The vascular ischemia associated with vasculitis caused atrophy of the stria vascularis and hair cell death, and, in the advanced stage, the progression of inflammation induced necrosis or cochlear fibrosis of the local tissue [44,45].(d)Abnormal reaction of cytotoxic T cells integrated into the inner ear [43]. In a presbycusis model mouse, dysfunctional T lymphocytes caused hearing loss at high frequencies, which was prevented by transplantation of normal bone marrow cells [40,43].

The inner ear lacks lymphoid tissue and has a blood inner ear barrier, which once made the inner ear an immunologically unique organ. However, recent studies have shown that the cells responsible for the local immune response are largely resident macrophages involved in maintaining inner ear homeostasis and immunity [36]. These immune cells play a key role in age-related or autoimmune hearing loss, so inner ear macrophages are expected to be important therapeutic targets [46]. 

## 7. Treatment for Sensorineural Hearing Loss in Sjögren’s Syndrome

It is clinically important that early diagnosis and early treatment are effective for SNHL [47,48]. In SLE, corticosteroid therapy is the most prevalent therapy and effective for SNHL [2,30], and additional or alternative therapies include the administration of anticoagulants [49], infliximab [50], and plasmapheresis [51]. 

Ciorba et al. [2] searched PubMed, Embase, and Cinahl databases for AIED over the past 10 years (January 2008 to December 2017), and, according to 50 articles, the main treatment for AIED was corticosteroids. AIED is one of the few SNHLs that is reversible, and prompt treatment may be beneficial. If treatment response and hearing improvement are observed at the end of 4 weeks of treatment, it is recommended that steroids be continued until clinical stability is achieved and the dose be reduced after approximately 6 months. Otherwise, in a non-responder patient, the steroid should be tapered over 10–14 days. Approximately 70% may respond to steroid therapy, but it may become less effective over time. The data from the literature estimate that the actual efficacy of steroid treatment was only 14%.

Green L et al. [49] reported that a 22 year-old man with anticardiolipin antibodies diagnosed with SLE failed to respond to treatment with prednisone for hearing loss but was successfully managed with low molecular weight heparin therapy followed by high-dose heparin and then coumadin.

Liu YC et al. [50] conducted a retrospective study to assess the effects of infliximab, a chimeric monoclonal antibody, on hearing in patients with autoimmune sensorineural hearing loss who had previously not responded to steroid therapy and/or treatment with other immunosuppressive drugs, such as methotrexate and cyclophosphamide. They reviewed the records of eight such patients, and no patient exhibited a positive response to infliximab therapy by objective measurements, and only one patient (12.5%) reported subjective improvement. However, all patients in this study were already refractory to all recommended therapies, including steroids, methotrexate, and cyclophosphamide. Therefore, the true efficacy of infliximab remains uncertain.

Luetje CM [51] reported successful plasmapheresis in a total of eight patients. The improvement of hearing disturbance was observed in six out of eight cases, and three of six cases did not require the immunosuppressant during follow-up over 3 years. These preliminary results suggest that plasmapheresis can stabilize or improve auditory and vestibular symptoms in selected patients.

For SNHL in SS patients, steroid treatment has only been reported to be effective during the early phase of hearing loss (Table 2).

## 8. Future Direction

In a 2021 study, Saito described the characteristics of McH/lpr-RA1 mice and proposed their use as a new mouse model for autoimmune sialadenitis [56]. MRL/lpr mice, which have been used as disease models for salivary gland inflammation, have a relatively short lifespan because they develop progressive nephritis. However, McH/lpr-RA1 mice are long-lived and display spontaneous autoimmune sialadenitis and submandibular vasculitis. Immunohistochemical studies discovered that salivary gland lesions strongly expressed four molecules associated with sialadenitis: SSA and SSB (Sjögren’s syndrome autoantigen) and gp91phox (promoter of reactive oxygen species production), single-stranded DNA (marker of apoptotic cells), indicating accelerated apoptosis of salivary gland cells triggered by oxidative stress. The inner ear may be damaged by the same mechanism as oxidative stress damaging salivary glands. Measurement of soluble gp91phox in serum, as a marker of oxidative stress, may be a potential indicator of glandular damage or inner ear disorders. Using the McH/lpr-RA1 mice, we could compare the levels of serum oxidative stress markers with the pathological images of inner ear disorders.

## 9. Strengths and Limitations of the Review

The strength of this article is that only a few case reports had described the treatment of patients with Sjögren’s syndrome and hearing loss, and our review article will be informative for clinicians who treat the patients. As a limitation of this article, basic research on the inner ear is essential to determine the cause of hearing loss in Sjogren’s syndrome, whereas only a small number of basic research papers have been published.

## 10. Conclusions

Underlying systemic autoimmunity, including SS, should always be considered in patients with vestibular symptoms, such as vertigo, tinnitus, and ear fullness or with progressive and variable hearing loss. The early diagnosis of AIED is crucial to avoid missing the opportunity to administer appropriate treatment. The development of molecular-targeted drugs for immune disorders is steadily progressing, and tissue macrophages in the inner ear may be potential therapeutic targets of hearing loss.

## Figures and Tables

**Figure 1 ijms-23-11181-f001:**
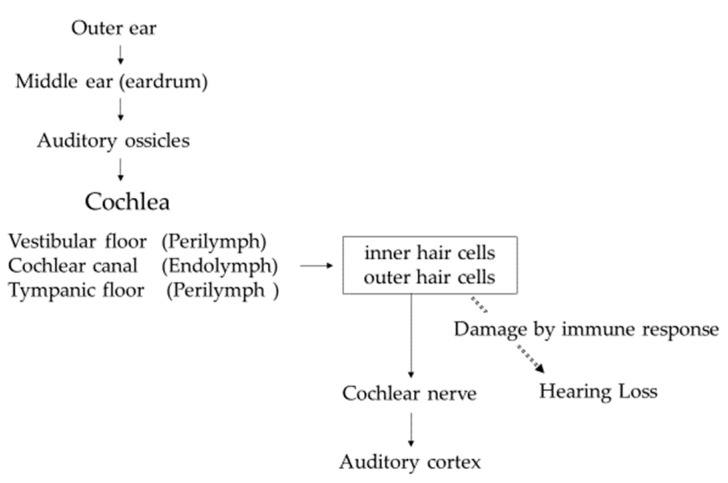
Sound transmission from outer ear to auditory cortex.

**Table 1 ijms-23-11181-t001:** Summary of the previous reports on SNHL in SS, RA, and SLE.

Reference [No.]	Disease	Number (Female/Male)	SNHL (%)	Frequency Band of SNHL
Tumiati B. et al. [28]	SS	30 (30/0)	46%	2000–8000 Hz (12 patients)
Ziavra N. et al. [29]	SS	45 (45/0)	22.5%	3000–8000 Hz (8 patients)
Galarza-Delgado DA. et al. [30]	SS	60 (60/0)	60% 70% 100%	500–3000 Hz 4000–8000 Hz 10,000–16,000 Hz
González JLT. et al. [31]	SS	63 (60/3)	95.2%	10,000–16,000 Hz
Gündüz B. et al. [32]	SS	36 (36/0)	52.77%	9000–12,500 Hz
Galarza-Delgado DA. et al. [30]	RA	117 (117/0)	36.8% 68.4% 94.9%	500–3000 Hz 4000–8000 Hz 10,000–16,000 Hz
Roverano S. et al. [10]	SLE	30 (30/0)	40% 13.3% 13.3%	2000–4000 Hz 4000–8000 Hz 2000–8000 Hz
Lasso de la Vega M. et al. [33]	SLE	77 (67/10)	70%	8000–18,000 Hz

PSL, prednisolone; mPSL, methylprednisolone.

**Table 2 ijms-23-11181-t002:** Summary of the case reports on treatment of Sjögren’s syndrome with SNHL.

Reference [No.]	Age (y)/Sex	Bilateral or Unilateral	Frequency Band (dB)	Treatment (Dose)	Effectiveness
Almeida RS. et al. [52]	65/Female	Unilateral	8000 Hz (70 dB)	PSL (1 mg/kg/day)	Effective, with recurrence
Kadosono O. et al. [53]	46/Female	Bilateral	ND	mPSL (500 mg/day for 3 days)	Effective
Kim KS. et al. [54]	62/Female	Unilateral	ND	mPSL (250 mg/day for 5 days)	Effective
Okawa Y. et al. [55]	8/Female	Unilateral	Deaf	PSL (1 mg/kg/day for 10 days, tapered off for 4 days)	Not effective
Okawa Y. et al. [55]	8/Male	Unilateral	Deaf	IVIG (1 g/kg) and PSL (1 mg/kg/day for 10 days, tapered off for 8 days)	Not effective

PSL, prednisolone; mPSL, methylprednisolone; IVIG, intravenous immunoglobulin; ND, not documented.

## Data Availability

Not applicable.

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
