# Peer review of "Sensorineural Hearing Loss in Sjögren’s Syndrome"

_ijms, 2022, doi:10.3390/ijms231911181_

Round 1
Reviewer 1 Report
This is a review of sensorineural hearing loss associated with Sjogren's Syndrome. It is not designed as a systematic review; rather it is an overview and describes reports of clinical findings and response to treatment with steroid.
Some of the paragraphs are not easy to read and the description of the cochlear mechanics could be improved, possibly by collaborating with an ORLHNS specialist.
strengths and limitations of the review and conclusions are not provided.
I think a minor revision could improve teh article.
Author Response
Response to Reviewer 1 Comments
Point 1: Some of the paragraphs are not easy to read and the description of the cochlear mechanics could be improved, possibly by collaborating with an ORLHNS specialist.
Response 1: Thank you for your advice. As you pointed out, the entire article of the revised manuscript was proofread as following the comments from an otolaryngology specialist.
Point 2: strengths and limitations of the review and conclusions are not provided.
Response 2: Thank you for your advice. We have added the strengths and limitations of this manuscript as follows.
9. Strengths and limitations of the review and conclusions
The strength of this article is that only a few case reports had described the treatment of patients with Sjögren's syndrome and hearing loss, and our review article will be informative for clinicians who treat the patients. As a limitation of this article, basic research on the inner ear is essential to determine the cause of hearing loss in Sjogren's syndrome, whereas only a small number of basic research papers have been published.
Reviewer 2 Report
Summary
This review paper summarises recent developments regarding the pathogenesis of sensorineural hearing loss associated with Sjögren's syndrome. The text describes the pathology of Sjögren's syndrome (SS), clinical manifestation of SS, Autoimmune inner ear disease, and histoanatomical features of the inner ear. The treatment for sensorineural hearing loss in SS is described, and a suggestion for future direction for research in this area is mentioned.
General comments
Overall, the review contains useful information, and will be of interest to readers of International Journal of Molecular Sciences. However, for a review article it is overly brief, and it is difficult to get an insight regarding exactly why the review is timely, what the gaps in the literature are, and how the summary of findings might be applied. The abstract does not give sufficient information regarding the content of the review itself, and the manuscript itself is lacking detail in places. I feel that these issues should be addressed in a revision that would strengthen the manuscript. I have some comments below, which I hope will be useful to the authors.
Specific comments
Abstract end: “We herein review recent developments concerning the pathogenesis of sensorineural hearing loss associated with Sjögren's syndrome.”
The majority of the abstract text describes Sjögren's syndrome, and sensorineural hearing loss. However, in the abstract it might help the reader to summarise the recent developments mentioned, so that they will have a better idea of the content of the review, rather than just having the descriptions of Sjögren's syndrome and sensorineural hearing loss.
Section 1: “Since early treatment with steroids has been reported to successfully minimalize the progression of hearing loss, the early detection of sensorineural hearing loss (SNHS) in the SS patients is considered important."
Please provide a supporting citation for this statement.
2-1. Prevalence of SS “The autoimmune disease SS is distributed worldwide, and its prevalence varies depending on ethnicity.”
This is rather vague, andsome numbers would help here. e.g. what is the estimated prevalence in different countries?
7. .Treatment for sensorineural hearing loss in SS
“It is clinically important that the early diagnosis and early treatment are effective for SNHL. In SLE, corticosteroid therapy is the most prevalent therapy and effective for SNHL [30], and additional or alternative therapies include the administration of anticoagulants [44], cyclophosphamide [45], monoclonal antibodies [46], and plasmapheresis [47]. For SNHL in SS patients, steroid treatment has only been reported to be effective during the early phase of hearing loss (Table B).”
This paragraph is overly brief. I would suggest that the studies that are cited are described in more detail.
8. Future direction “In their 2021 study, Saito [48] described the characteristics of McH/lpr-RA1 mice and proposed their use as a novel murine model of autoimmune sialadenitis. This new model mouse has a longer life than MRL/lpr mice, which have been used as a disease model for salivary gland inflammation but have a relatively short life span because of the development of progressive nephritis. Further basic research using animal models will elucidate the pathophysiology of SS.”
This is too brief. What other future directions might research take in this area? What is still unknown that needs to be addressed? “Further basic research using animal models will elucidate the pathophysiology of SS” is a rather vague statement, and some more detail regarding why this is important, or exactly how such work will elucidate the pathophysiology of SS would be helpful.
Author Response
Response to Reviewer 2 Comments
Point 1: Abstract end: “We herein review recent developments concerning the pathogenesis of sensorineural hearing loss associated with Sjögren's syndrome.”
The majority of the abstract text describes Sjögren's syndrome, and sensorineural hearing loss. However, in the abstract it might help the reader to summarise the recent developments mentioned, so that they will have a better idea of the content of the review, rather than just having the descriptions of Sjögren's syndrome and sensorineural hearing loss.
Response 1: Thank you for your advice. I corrected the abstract as follows.
This review article consists of (1) Pathology of Sjögren's syndrome (SS), (2) Clinical manifestation of SS, (3) Autoimmune inner ear disease (AIED), (4) Histoanatomical features of the inner ear, (5) Immunological characteristics in the inner ear, (6) Pathophysiology of AIED, (7) Treatment for sensorineural hearing loss in SS, and (8) Future direction. Finally, we introduce a recently developed disease model of salivary gland inflammation and discuss future expectations for the treatment of sensorineural hearing loss in SS.
Point 2: Section 1: “Since early treatment with steroids has been reported to successfully minimalize the progression of hearing loss, the early detection of sensorineural hearing loss (SNHS) in the SS patients is considered important."
Please provide a supporting citation for this statement.
Response 2: Thank you for your advice. Reference numbers [57] have been added in the text.
Point 3: 2-1. Prevalence of SS “The autoimmune disease SS is distributed worldwide, and its prevalence varies depending on ethnicity.”
This is rather vague, and some numbers would help here. e.g. what is the estimated prevalence in different countries?
Response 3: Thank you for your advice. We apologize for the misleading expression "prevalence". We meant to show the frequency of clinical symptoms and laboratory data according to specific characteristics such as ethnicity. Therefore, I added the following sentences:
The Big Data Sjögren Project Consortium conducted a multivariate logistic regression analysis involving 20 centers from 5 continents by January 2016 on an international multi-center registry designed in 2014. An analysis of this registry [2] included 7748 women (93%) and 562 men (7%), with a mean age at diagnosis of primary SjS of 53 years old, 95% of which are available for ethnicity. This study demonstrated that SjS was diagnosed on average 7 years earlier in black/African Americans compared with white patients. The female to male ratio was highest in Asian patients (27:1) and lowest in black/African American patients (7:1).
Point 4:
- Treatment for sensorineural hearing loss in SS
“It is clinically important that the early diagnosis and early treatment are effective for SNHL. In SLE, corticosteroid therapy is the most prevalent therapy and effective for SNHL [30], and additional or alternative therapies include the administration of anticoagulants [44], cyclophosphamide [45], monoclonal antibodies [46], and plasmapheresis [47]. For SNHL in SS patients, steroid treatment has only been reported to be effective during the early phase of hearing loss (Table B).”
This paragraph is overly brief. I would suggest that the studies that are cited are described in more detail.
Response 4:
Thank you for your advice. I've added more detail to the actual treatment as follows.
Ciorba [57] et al. searched PubMed, Embase, and Cinahl databases for AIED over the past 10 years (January 2008 to December 2017), and according to the 50 articles, the main treatment for AIED was corticosteroids. AIED is one of the few SNHLs that is reversible and prompt treatment may be beneficial. If treatment response and hearing improvement are observed at the end of 4 weeks of treatment, it is recommended that steroids be continued until clinical stability is achieved and the dose be reduced after approximately 6 months. Otherwise, in a non-responder patient, the steroid should be tapered over 10-14 days. Approximately 70% may respond to steroid therapy, but it may become less effective over time. The data from the literature estimate that the actual efficacy of steroid treatment was only 14%.
Green L [44] et al. reported that a 22 year-old man with anticardiolipin antibodies diagnosed with SLE failed to treatment with prednisone for hearing loss but was successfully managed with low molecular weight heparin therapy followed by high-dose heparin and then coumadin.
Liu YC [46] et al. conducted a retrospective study to assess the effects of infliximab, a chimeric monoclonal antibody, on hearing in patients with autoimmune sensorineural hearing loss who had previously not responded to steroid therapy and/or treatment with other immunosuppressive drugs such as methotrexate and cyclophosphamide. They reviewed the records of 8 such patients and no patient exhibited a positive response to infliximab therapy by objective measurements, and only 1 patient (12.5%) reported subjective improvement. However, all patients in this study were already refractory to all recommended therapies, including steroids, methotrexate, and cyclophosphamide. Therefore, the true efficacy of infliximab remains uncertain.
Luetje CM [47] reported successful plasmapheresis in a total of 8 patients. The improvement of hearing disturbance was observed in 6 out of 8 cases, and 3 of 6 cases did not require the immunosuppressant during followed-up over 3 years. These preliminary results suggest that plasmapheresis can stabilize or improve auditory and vestibular symptoms in selected patients.
Point 5:
- Future direction “In their 2021 study, Saito [48] described the characteristics of McH/lpr-RA1 mice and proposed their use as a novel murine model of autoimmune sialadenitis. This new model mouse has a longer life than MRL/lpr mice, which have been used as a disease model for salivary gland inflammation but have a relatively short life span because of the development of progressive nephritis. Further basic research using animal models will elucidate the pathophysiology of SS.”
This is too brief. What other future directions might research take in this area? What is still unknown that needs to be addressed? “Further basic research using animal models will elucidate the pathophysiology of SS” is a rather vague statement, and some more detail regarding why this is important, or exactly how such work will elucidate the pathophysiology of SS would be helpful.
Response 5: Thank you for your advice. About Future direction, I wrote a lot as follows.
In a 2021 study, Saito [48] described the characteristics of McH/lpr-RA1 mice and proposed their use as a new mouse model for autoimmune sialadenitis. MRL/lpr mice, which have been used as disease models for salivary gland inflammation, but have a relatively short lifespan because they develop progressive nephritis. However, McH/lpr-RA1 mice are long-lived and display spontaneous autoimmune sialadenitis and submandibular vasculitis. Immunohistochemical studies discovered that salivary gland lesions strongly expressed four molecules associated with sialadenitis: SSA and SSB (Sjögren's syndrome autoantigen) and gp91phox (promoter of reactive oxygen species production), single-stranded DNA (marker of apoptotic cells), indicating accelerated apoptosis of salivary gland cells triggered by oxidative stress. The inner ear may be damaged by the same mechanism as oxidative stress damaging salivary glands. Measurement of soluble gp91phox in serum, as a marker of oxidative stress, may be a potential indicator of glandular damage or inner ear disorders. Using the McH/lpr-RA1 mice, we could compare the levels of serum oxidative stress markers with the pathological images of inner ear disorders.
Round 2
Reviewer 1 Report
The article reads better, thank you. The references are not listed sequentially as they appear in the text. Some of the references listed do not have cross-reference to your text. The complete reference list requires revision and reordering.
Once completed it should be ready for publication
Author Response
Response to Reviewer 1 Comments
Point 1: The article reads better, thank you. The references are not listed sequentially as they appear in the text. Some of the references listed do not have cross-reference to your text. The complete reference list requires revision and reordering.
Once completed it should be ready for publication.
Response 1: Thank you for your comment. We have re-numbered the references appropriately. Thank you for your kind review and advice.
Reviewer 2 Report
I appreciate the efforts of the authors in addressing the comments from the last round of reviews. The updated manuscript indicates that the previous points have been dealt with appropriately. I have no further concerns.
Author Response
Response to Reviewer 2 Comments
Point 1: I appreciate the efforts of the authors in addressing the comments from the last round of reviews. The updated manuscript indicates that the previous points have been dealt with appropriately. I have no further concerns.
Response 1:
Thank you very much for your kind review and advice during your busy schedule. Thank you very much.